# TRIP: Accelerating Document-level Multilingual Pre-training via Triangular Document-level Pretraining on Parallel Data Triplets[*]

**Hongyuan Lu**[♡†], **Haoyang Huang**[♠], **Shuming Ma**[♠], **Dongdong Zhang**[♠],
**Wai Lam**[♡], **Zhaochuan Gao**[♠], **Anthony Aue**[♠], **Arul Menezes**[♠], **Furu Wei**[♠]
[♡]The Chinese University of Hong Kong
[♠]Microsoft Corporation
{hylu,wlam}@se.cuhk.edu.hk
{haohua,shumma,dozhang,anthaue,arulm,fuwei}@microsoft.com

## Abstract

Despite the success of multilingual sequence-to-sequence pre-training, most existing approaches rely on document-level monolingual corpora in many different languages, sentence-level bilingual corpora,[1] and sometimes synthetic document-level bilingual corpora. This hampers the performance with cross-lingual document-level tasks such as document-level translation. Hence, we propose to mine and leverage document-level trilingual parallel corpora to improve sequence-to-sequence multilingual pre-training. We present **Tri**angular Document-level **P**re-training (**TRIP**) as the first in the field to accelerate the conventional monolingual and bilingual objectives into a trilingual objective with a novel method called Grafting. Experiments show that TRIP achieves several strong state-of-the-art (SOTA) scores on three multilingual document-level machine translation benchmarks and one cross-lingual abstractive summarization benchmark, including consistent improvements by up to 3.11 d-BLEU points and 8.9 ROUGE-L points.

## 1 Introduction

Conventional multilingual pre-training achieved promising results on machine translation (Liu et al., 2020) and cross-lingual classification (Xue et al., 2021). These pre-training paradigms usually rely on monolingual corpora in many different languages, with denoising objectives such as sentence permutation and span masking (Liu et al., 2020;

Lewis et al., 2020b). Following the calls that the unsupervised scenario is not strictly realistic for cross-lingual learning (Artetxe et al., 2020), multilingual pre-training advanced into a supervised setting through sentence-level bilingual translation pairs (Chi et al., 2021; Reid and Artetxe, 2022) to provide a stronger signal for pre-training. Among these pioneering works, document-level multilingual pre-training with parallel data is currently an understudied topic. This direction is particularly significant for tasks that necessitate contextual comprehension, such as document-level machine translation and cross-lingual summarization. As a workaround, DOCmT5 (Lee et al., 2022) resorts to using synthetic bilingual translation pairs to scale up document-level multilingual pre-training.

In addition to the lack of study for document-level multilingual pre-training with parallel data, prior works also overlooked the value of trilingual parallel data for multilingual pre-training. Compared to bilingual parallel data, trilingual parallel data is expected to better capture different linguistic clues and coherence among different languages such as past tense and gendered expressions,[2] which can enhance the model pre-training on aspects of document-level cross-lingual understanding and resolve cross-lingual ambiguities.

To this end, we present TRIP, a document-level multilingual pre-training method using trilingual parallel corpora. Because there is no publicly available document-level trilingual corpus, we propose a novel method to construct trilingual document pairs from document-level bilingual corpora. Subsequently, we augment the conventional multilingual pre-training by (i) Grafting two documents presented in two different languages into one mixed document, and (ii) predicting the remaining one language as the reference translation.

---

[*]The work described in this paper is substantially supported by a grant from the Research Grant Council of the Hong Kong Special Administrative Region, China (Project Code: 14200719).

[†]Contribution during an internship at Microsoft Research Asia.

[1]In this paper, we use 'bilingual corpora' to denote parallel corpora with 'bilingual translation pairs' in many different language pairs, each consisting of two sentences/documents with the same meaning written in different languages. We use 'trilingual corpora' to denote parallel corpora with 'trilingual translation pairs' in many different language combinations, each consisting of three sentences/documents.

---

[2]For example, Chinese does not have past tense for verbs, while Japanese and English do have relevant clues. See Figure 1 for further explanation.

| Models | Denoising Pre-training | Translation Pre-training | Trilingual Document Pairs | Trilingual Objective | Document Level |
|---|---|---|---|---|---|
| mBART | ✓ | ✗ | ✗ | ✗ | ✓ |
| mT5 | ○ | ✗ | ✗ | ✗ | ✓ |
| mT6 | ○ | ✓ | ✗ | ✗ | ✓ |
| PARADISE | ○ | ✓ | ✗ | ✗ | ✗ |
| DOCmT5 | ✓ | ✓ | ✗ | ✗ | ✓ |
| **TRIP** | ✓ | ✓ | ✓ | ✓ | ✓ |

Table 1: **Comparisons of various multilingual pre-training methods.** We denote the intermediate value as ○. For example, mT5 uses span corruption solely without sentence permutation, so we put a value of ○ for the column of Denoising Pre-training for mT5. The columns of **Denoising Pre-training** and **Translation Pre-training** refer to the pre-training objectives we introduce at the start of Section 2.

We conduct experiments on document-level machine translation on TED Talks (Cettolo et al., 2015), News benchmark (News-commentary) and Europarl (Koehn, 2005), and cross-lingual abstractive summarization on Wikilingua (Ladhak et al., 2020; Gehrmann et al., 2021). We found that TRIP clearly improves previous multilingual pre-training paradigms that use monolingual and bilingual objectives (Lee et al., 2022), and achieves strong SOTA results on both tasks.

In summary, we make three key contributions:

- TRIP proposes a novel trilingual pre-training objective through Grafting for multilingual pre-training, along with a novel method to construct trilingual data from parallel corpora.

- TRIP yields SOTA scores on both multilingual document-level MT and cross-lingual abstractive summarization.

- We conduct in-depth analyses on document-level cross-lingual understanding and compare TRIP to commercial systems.

## 2 Triangular Document-level Pre-training

We start by introducing the conventional methodologies previously used by the monolingual and bilingual objectives for multilingual pre-training:

- **Denoising Pre-training**: Sentence permutation (Liu et al., 2020) and span corruption (Xue et al., 2021) are effective denoising pre-training objectives for document-level multilingual pre-training.

- **Translation Pre-training**: Making the use of sentence-level translation pairs is a bilingual pre-training strategy for multilingual models (Kale et al., 2021; Tang et al., 2021).

**Constructing a Trilingual Objective** In comparison, **TRIP** is the first in the field to introduce a trilingual objective for multilingual pre-training. The core to making better use of trilingual data is to **Grafting**[3] the documents by splitting the documents written in two different languages but with the same meaning half by half and concatenating each half to form a new document that retains the same meaning written in two different languages. TRIP then applies sentence permutation and span corruption on the Grafted documents.

Conventional monolingual and bilingual pre-training objectives overlooked the value to take such an advantage (Liu et al., 2020; Reid and Artetxe, 2022) of linguistic clues from different languages. In contrast, TRIP fuses authentic trilingual data, in which linguistic clues such as past tense and gendered nouns are usually preserved.

We present in Figure 1 to illustrate how TRIP operates to make use of linguistic clues through trilingual data. Given three documents with the same meaning written in Chinese, Japanese, and English, two of the documents are split and concatenated. The concatenation is randomly permutated at the sentence level, and the remaining unchanged document is used as the translation reference. Here, Chinese is tenseless, and TRIP effectively fuses useful linguistic clues for past tense written in Japanese and English into the Chinese text to resolve cross-lingual ambiguities.

Table 1 presents the characteristics that TRIP exhibits compared to existing methods. We report whether the models use trilingual document pairs for pre-training, and we report whether document-level tasks such as document-level machine translation or abstractive summarization are reported in their original papers. To our best knowledge, this is the first paper in our field to mine and use trilingual document pairs for multilingual pre-training. This is also the first work that features Grafting.

More formally, we first denote $\mathcal{N}$ as the number of training document pairs in trilingual translation triplets of $(x_1, x_2, x_3)$ in a pre-training corpus $\mathcal{D}$. Given a Seq2Seq generation model (Sutskever

---

[3]Grafting refers to joining two plants together by cutting and using scion (the upper part of the grafting) as the top and the understock (the lower part of the grafting) as the root.

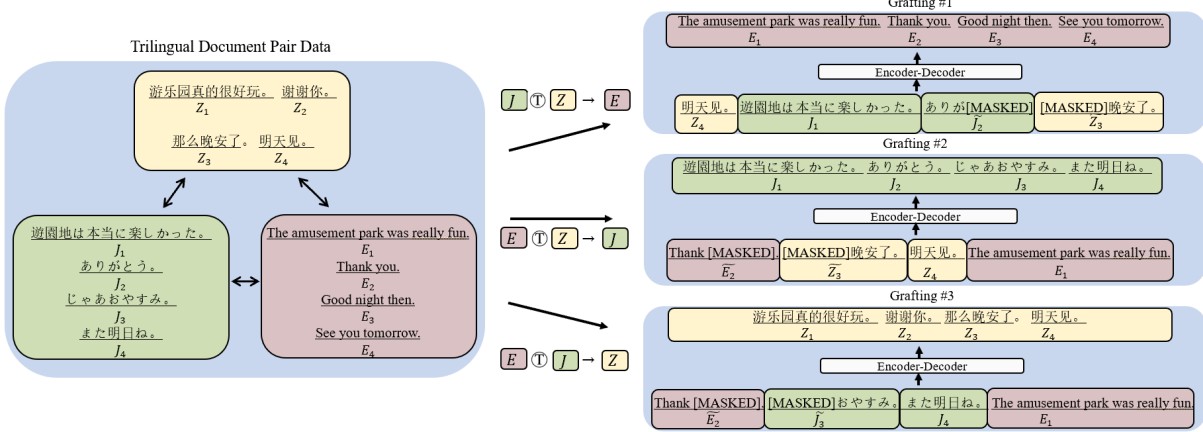

Figure 1: Overview of **Tri**angular Document-level **P**re-training (**TRIP**). We select three languages for demonstration. Here, Chinese is tenseless, and Japanese and English contain past tense as the linguistic clues that can resolve cross-lingual ambiguities. We present three Grafting cases, representing three different language combinations. For each trilingual pair, two languages serve as the input with the remaining one as the reference translation. We define a novel symbol ⓣ that denotes a noise function that combines operations in sequence: splitting by half and concatenation (**Grafting**), and sentence permutation and span corruption. $Z_n$, $J_n$, and $E_n$ for $n = \{1, 2, 3, 4\}$ denotes four sentences written in three different languages. $\tilde{Z}_n$, $\tilde{J}_n$, and $\tilde{E}_n$ denotes corrupted sentences.

et al., 2014), TRIP optimizes the likelihood:

$$\sum_{n=1}^{\mathcal{N}} \mathbb{E}_{x_1^n, x_2^n, x_3^n \in \mathcal{D}}[-\log P_\theta(x_3 \mid x_1 \text{ ⓣ } x_2)], \quad (1)$$

where we define ⓣ as a novel operation that takes two documents in different languages as the input and takes three operations in sequence: splitting by half, concatenating, and sentence permutation.[4]

**Creating Trilingual Document Pairs** As there is no public corpus with trilingual document pairs, TRIP creates **MTDD** (**M**icrosoft **T**rilingual **D**ocument **D**ataset), a high-quality trilingual parallel corpus with document translation pairs across 67 languages, 4,422 bilingual directions, and 99,628 trilingual combinations. The corpus is sourced from high-quality news documents scoped from an in-house website[5] timestamped from April 2021 to July 2022. The whole procedure is composed of two steps: (i) creating bilingual document pairs and (ii) creating trilingual document pairs based on the bilingual document pairs.

To obtain bilingual document pairs, we follow ParaCrawl (Bañón et al., 2020) to translate all the documents we have into English using a light-weighted word-based machine translation model.

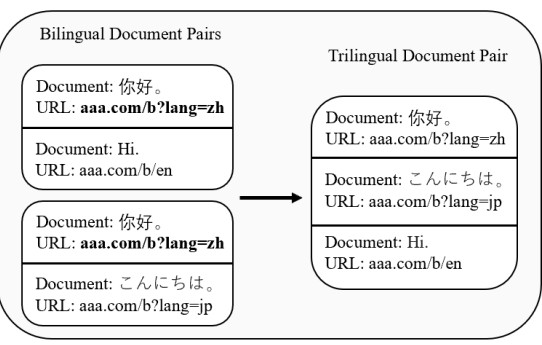

Figure 2: Illustration for the URL matching mechanism to create trilingual document pairs from bilingual data. In this case, we construct trilingual data by successfully matching the URL address for the Chinese document.

The resulting translation is used for pairing only and the documents are paired and thresholded with similarity scores such as tf-idf computed on their English translation (Bañón et al., 2020). To improve efficiency, we attempt to pair documents only if they are timestamped within a small window such as one week. The motivation is that the semantic news with the same meaning in different languages are often reported within a small timestamp window in high probabilities. The resulting document pairs are further thresholded and filtered with LASER (Artetxe and Schwenk, 2018),[6] which is a multilingual sentence representation.

Given the bilingual data constructed as above,

---

[4]As a pre-training method, TRIP is robust and does not require the sentences in trilingual document pairs to be perfectly aligned in their orderings. Filtering the non-perfect pairs can throw away the data and deteriorate the performance gains.

[5]www.bing.com

[6]https://github.com/facebookresearch/LASER

we follow previous works (Bañón et al., 2020; El-Kishky et al., 2020). These previous works leverage URL addresses for constructing bilingual data. In contrast, we use URL addresses to construct trilingual data pairs by matching and linking. Figure 2 depicts a detailed illustration.

For space reasons, we present statistics to illustrate the scale of MTDD in Table 10 in Appendix D. We also note that existing MTmC4 (Lee et al., 2022) used by DOCmT5 can be less favourable for our experiments as (i) MTmC4 is composed of synthetic data that could be of lower quality, (ii) MTmC4 is not publicly available at the time of writing, and (iii) MTmC4 can lead to potential data leakage for the test sets on TED Talks.

## 3 Experiments

### 3.1 TRIP Pre-training

**Model Configuration** We use a Transformer architecture that is composed of 24 Transformer encoder layers and 12 interleaved decoder layers. In addition, it has an embedding size of 1024, and a dropout rate of 0.1. The feed-forward network is configured to have a size of 4096 with 16 attention heads. For parameter initialization, we follow Ma et al. (2021) and Yang et al. (2021) to train a sentence-level MT system. The motivation is that previous studies have shown that the hybrid training of sentence-level and document-level MT can improve the performance of document-level translation (Sun et al., 2022). We call it the Baseline Model in the remaining of this paper.

**Data and Pre-processing** As described in Section 2, we create a trilingual document-level corpus, MTDD, for TRIP pre-training with the use of trilingual document pairs. We create a list of keywords to automatically clean and remove noisy text such as claims and advertisements. We follow Ma et al. (2021) to use SentencePiece (Kudo and Richardson, 2018) for tokenization, and we use the same SentencePiece model as Yang et al. (2021). Following the previous works, we prefix the inputs with a language tag that indicate the target language of the generation for both pre-training and fine-tuning.

**Training Details** We use the Adam optimizer (Kingma and Ba, 2014) with $\beta_1 = 0.9$ and $\beta_2 = 0.98$ for our multilingual pre-training. The learning rate is set as $1e$-5 with a warmup step of 4000. We use the label smoothing cross-entropy for our translation loss and we set label smoothing with a ratio

| Model | Fr→En | De→En | Zh→En | Vi→En | Cs→En | Th→En | Avg. |
|---|---|---|---|---|---|---|---|
| *Sentence-level MT Models* | | | | | | | |
| HAN† | - | - | 24.00 | - | - | - | - |
| M2M-100 | 50.18 | 42.24 | 26.62 | 34.92 | 37.84 | 27.28 | 36.51 |
| mBART | 48.69 | 44.80 | 28.39 | 37.18 | 39.47 | - | - |
| Baseline Model | 50.69 | 47.07 | 30.35 | 39.59 | 43.05 | 32.30 | 40.51 |
| *Document-level MT Models* | | | | | | | |
| mT5† | - | - | 24.24 | - | - | - | - |
| M2M-100 | 49.43 | 43.82 | 26.63 | 35.91 | 39.04 | 25.93 | 36.79 |
| mBART | 49.16 | 44.86 | 29.60 | 37.09 | 39.64 | - | - |
| MARGE† | - | - | 28.40 | - | - | - | - |
| DOCmT5† | - | - | 31.40 | - | - | - | - |
| Baseline Model | 49.53 | 45.98 | 30.17 | 39.28 | 42.33 | 30.62 | 39.65 |
| Baseline Model$^+$ | 50.74 | 46.46 | 30.65 | 39.67 | 42.64 | 31.70 | 40.31 |
| **TRIP (Ours)** | **51.94** | **48.24** | **31.63** | **40.52** | **44.22** | **32.87** | **41.57** |

Table 2: Results for document-level MT on TED Talks in the direction of (X → En). We report the d-BLEU scores for all the results. †: scores are taken from the official papers for these models. -: the scores are not reported or the language is not supported. The Baseline Model refers to the model described in Section 3.1. The Baseline Model$^+$ represents a document-level model continually pre-trained with the bilingual data in MTDD. For a fair comparison, the trilingual data used by TRIP are constructed from these bilingual data. We perturbed them on sentence permutation and span corruption as the noise functions, with no use of trilingual data.

of 0.1 for model training. All of our pre-trainings are conducted on 16 NVIDIA V100 GPUs. We set the batch size as 512 tokens per GPU. To simulate a larger batch size, we update the model every 128 steps. For the Grafting operation Ⓣ defined for TRIP, we split the documents 50% by 50%.

### 3.2 Multilingual Document-level MT

#### 3.2.1 TED Talks

**Experimental Settings** Following DOCmT5, we use the IWSLT15 Campaign for the evaluation of TED Talks. Prior systems have reported scores on only 1 or 2 translation directions (Lee et al., 2022; Sun et al., 2022), and DOCmT5 supports only the translation direction into English (X → En). We report more language directions while DOCmT5 only evaluates on (Zh → En). Following DOCmT5, we split all documents into a maximum of 512 tokens for all train/dev/test sets during training and inference. We use the official parallel training data from IWSLT15 without any additional monolingual data, with the official 2010 dev set and 2010-2013 test set for evaluation (Lee et al., 2022). We compute d-BLEU (Papineni et al., 2002; Liu et al., 2020; Bao et al., 2021), a BLEU score for documents.

| Model | Fr→En | De→En | Zh→En | Cs →En | Avg. |
|---|---|---|---|---|---|
| *Sentence-level MT Models* | | | | | |
| M2M-100 | 31.58 | 25.65 | 18.47 | 28.17 | 25.97 |
| mBART | 29.93 | 29.31 | 18.33 | 30.15 | 26.93 |
| Baseline Model | 35.59 | 34.71 | 27.23 | 37.39 | 33.73 |
| *Document-level MT Models* | | | | | |
| M2M-100 | 32.67 | 25.78 | 17.85 | 29.06 | 26.34 |
| mBART | 30.14 | 26.35 | 15.01 | 29.79 | 25.32 |
| Baseline Model | 36.38 | 34.24 | 25.58 | 36.97 | 33.29 |
| Baseline Model$^+$ | 38.47 | 35.20 | 26.74 | 37.26 | 34.42 |
| **TRIP (Ours)** | **39.49** | **35.48** | **27.58** | **38.06** | **35.15** |

Table 3: Results for document-level MT on the News benchmark in the direction of (X → En).

We use SacreBLEU for evaluation.[7]

**Baseline Systems**  We report strong baselines evaluated at both sentence and document levels, including SOTA models DOCmT5† (Lee et al., 2022), M2M-100 (Fan et al., 2022), mBART (Liu et al., 2020), HAN† (Miculicich et al., 2018), MARGE† (Lewis et al., 2020a), and the Baseline Model that we use to initialize the weights for TRIP. †: the scores are taken from existing papers. We also compare to the Baseline Model$^+$, a document-level model pre-trained continually on the Baseline Model with the bilingual data used to construct the trilingual data in MTDD. We do not compare to PARADISE (Reid and Artetxe, 2022), a pre-trained model that uses dictionary denoising on monolingual data, as its weights are not publicly available so far. During our trials, we found that monolingual dictionary denoising can degrade document-level systems. We think that it could better serve sentence-level tasks such as sentence-level MT and cross-lingual classification as conducted in its original paper. See Appendix C for the number of model parameters.

**Results**  Table 2 presents the evaluation results for TED Talks in the directions of (X → En). TRIP clearly surpasses the baselines. TRIP surpasses the Baseline Model when both are fine-tuned at the document level by an average of 1.87 points in d-BLEU. TRIP surpasses the Baseline Model fine-tuned at the sentence level by an average of 1.01 points in d-BLEU. We postulate that the Baseline Model fine-tuned at the document level is no better than that of the sentence level due to the reason of the long input problem (Koehn and Knowles, 2017), and also due to the reason that the Baseline Model itself is pre-trained at the sentence level. TRIP

----
[7]https://github.com/mjpost/sacrebleu

beats the prior SOTA system DOCmT5. For space reasons, we present in Appendix A the evaluations in the (X→X) directions, which also show that TRIP effectively improves language pairs that are unseen during pre-training.

We also found that (i) the Baseline Model$^+$ clearly surpasses the Baseline Model and (ii) TRIP clearly surpasses the Baseline Model$^+$. This observation indicates two points: (i) the bilingual data in MTDD used to construct the trilingual data are of high quality and (ii) the trilingual objective with the Grafting mechanism is superior to the conventional bilingual objectives for multilingual pre-training.

### 3.2.2 News

**Experimental Settings**  For evaluation on the News benchmark, we follow Sun et al. (2022) to use News Commentary v11 as the training set. For Cs and De, we use newstest2015 as the dev set, and newstest2016/newstest2019 as the test set respectively. For Fr, we use newstest2013 as the dev set and newstest2015 as the test set. For Zh, we use newstest2019 as the dev set and newstest2020 as the test set. We use the same dataset preprocessing and evaluation metric as for the TED Talks.

**Baseline Systems**  As the weights for DOCmT5 are not available at the time of writing, we compare our system to various strong baselines such as M2M-100, mBART, the Baseline Model, and the Baseline Mode$^+$. The scores are obtained by fine-tuning the official checkpoints.

**Results**  Table 3 shows obvious and consistent improvements by up to 3.11 d-BLEU points (from 36.38 to 39.49) with TRIP for (Fr → En) compared to the Baseline Model.

### 3.2.3 Europarl

**Experimental Settings**  For the Europarl dataset (Koehn, 2005), we follow Sun et al. (2022) to use Europarl-v7, and we experiment with the setting of (X → En) where we test nine languages: Da, De, El, Es, Fr, It, Nl, Pt, and Sv. Like previous works (Bao et al., 2021; Sun et al., 2022), the dataset is randomly partitioned into train/dev/test divisions. Additionally, we split by English document IDs to avoid information leakage.

**Baseline Systems**  As the weights for DOCmT5 are not available at the time of writing, we compare our system to various strong baselines such as M2M-100, mBART, the Baseline Model, and

| Model | Da→En | De→En | El→En | Es→En | Fr→En | It→En | Nl→En | Pt→En | Sv→En |
|---|---|---|---|---|---|---|---|---|---|
| | | | | *Sentence-level MT Models* | | | | | |
| M2M-100 | 50.40 | 47.38 | 52.28 | 52.03 | 48.26 | 49.70 | 46.78 | 49.84 | 52.34 |
| mBART | - | 48.28 | - | - | 49.16 | 50.83 | 47.48 | - | - |
| Baseline Model | 48.94 | 47.25 | 53.46 | 50.57 | 47.68 | 49.49 | 45.95 | 50.65 | 52.77 |
| | | | | *Document-level MT Models* | | | | | |
| M2M-100 | 50.33 | 47.00 | 52.24 | 52.14 | 48.13 | 49.71 | 46.65 | 40.68 | 52.28 |
| mBART | - | 47.70 | - | - | 48.98 | 50.62 | 46.96 | - | - |
| Baseline Model | 49.85 | 47.64 | 53.34 | 51.32 | 48.46 | 50.26 | 47.12 | 50.13 | 52.42 |
| Baseline Model[+] | 49.90 | 47.75 | 53.75 | 51.78 | 48.70 | 50.37 | 47.18 | 50.49 | 52.49 |
| **TRIP (Ours)** | **51.13** | **48.30** | **54.38** | **52.29** | **49.36** | **51.23** | **48.07** | **51.03** | **53.43** |

Table 4: Results for document-level machine translation on Europarl in the direction of (X → En).

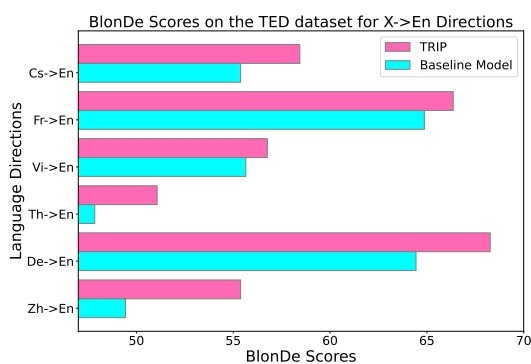

Figure 3: BlonDe scores on the TED Talks evaluated with TRIP and the Baseline Model (Document-level).

the Baseline Model[+]. The scores are obtained by fine-tuning the official checkpoints.

**Results**    By comparing TRIP to strong baselines, we see that the improvements with TRIP are consistent in all directions, and surpass all the strong baselines. This validates TRIP's effectiveness.

### 3.2.4    Coherence and Consistency Evaluation

**BlonDe Evaluation**    Figure 3 depicts the evaluations on TED Talks with BlonDe scores (Jiang et al., 2022), an evaluation metric designed for document-level MT which considers coherence and consistency issues that require the model to resolve cross-lingual ambiguities. Consistent improvements can be observed in all the directions on TED Talks with TRIP, meaning that TRIP generates more coherent and consistent translations than the baseline does. As discussed in Section 2, we postulate that these improvements attribute to the Grafting mechanism that resolves cross-lingual ambiguities by exploiting useful linguistic clues in trilingual data. This improves translation in coherence and consistency as reflected in the BlonDe scores. We demonstrate case studies for more analysis of coherence and consistency issues.

**Case Study**    Table 5 presents three case studies that demonstrate and compare the outputs between TRIP and the baseline systems. We highlight the correct translation in aqua and the wrong translation in hot pink. In addition to the comparison to the Baseline Models, we also present the outputs from popular commercial translation systems Google Translate, Microsoft Translator, and DeepL Translate. Each case demonstrates that TRIP is the best in terms of three characteristics respectively: (i) tense consistency (Jiang et al., 2022; Sun et al., 2022) across the sentences, (ii) noun-related issues (Jiang et al., 2022) such as singular and plural consistency as well as attaching definite article 'the' to a previously mentioned object 'light', and (iii) conjunction presence that indicates the relationship between sentences and makes the translation natural and fluent (Xiong et al., 2019; Sun et al., 2022). While some translations in the third case are acceptable, missing coordinating conjunction does not precisely capture the relationship between sentences and can make the translation less fluent.

TRIP is the best one among the systems at resolving cross-lingual ambiguities. This observation highlights the necessity of translating with document-level contexts for resolving cross-lingual ambiguities. The observations also align with the BlonDe measurements reported above.

### 3.2.5    Large Language Models

Table 7 compares TRIP to popular ChatGPT (GPT-3.5-TURBO)[8] on TED Talks. We use a prompt: "Translate the following text into English:". The results indicate ChatGPT still lags behind supervised system TRIP on document-level MT. This conclusion aligns with the previous study on sentence-level MT (Zhu et al., 2023), and we postulate that the reason is ChatGPT fails in handling contexts perfectly for document-level MT.

---

[8]https://chat.openai.com/chat

| | Case 1: Tense Consistency (Jiang et al., 2022; Sun et al., 2022) |
|---|---|
| Source | ……，但是这是一个大致的抽象的讨论，当某些间隙的时候，奥克塔维奥说，"保罗,也许我们可以观看TEDTalk。" TEDTalk用简单的方式就讲明了，…… |
| Reference | ..., But it was a fairly abstract discussion, and at some point when there was a pause, Octavio said, "Paul, maybe we could watch the TEDTalk." So the TEDTalk laid out in very simple terms, ... |
| Google Translate | ..., But it's a roughly abstract discussion when at some point Octavio said, "Paul, maybe we can watch the TEDTalk." The TEDTalk said it in a simple way, ... |
| Microsoft Translator | ..., But it's a roughly abstract discussion when, at certain intervals, Octavio said, "Paul, maybe we can watch TEDTalk." TEDTalk explains it in a simple way, ... |
| DeepL Translate | ..., But it was a broadly abstract discussion, and when there were certain breaks, Octavio said, "Paul, maybe we can watch TEDTalk." TEDTalk uses simple way to illustrate, ... |
| Baseline Model (Sentence-level) | ..., But it's kind of an abstract discussion, and at some point, Octavio says, "Paul, maybe we can watch the TEDTalk." And the TEDTalk simply explains that, ... |
| Baseline Model (Document-level) | ..., But it's sort of an abstract discussion. And at some point, Octavio said, "Paul, maybe we can watch the TEDTalk." The TEDTalk explained, in a very simple way, ... |
| TRIP | ..., But it was a sort of abstract discussion, and at some point in the intermission, Octavio said, "Paul, maybe we can watch the TEDTalk." And the TEDTalk made it clear, ... |
| | Case 2: Noun-related Issues (Jiang et al., 2022) |
| Source | ……，当光在西红柿上走过时，它一直在闪耀。它并没有变暗。为什么？因为西红柿熟了，并且光在西红柿内部反射，…… |
| Reference | ..., as the light washes over the tomato, It continues to glow. It doesn't become dark. Why is that? Because the tomato is actually ripe, and the light is bouncing around inside the tomato, ... |
| Google Translate | ..., as the light passed over the tomatoes, It kept shining. It didn't get darker. Why? Because the tomatoes are ripe, and light is reflected inside the tomatoes, ... |
| Microsoft Translator | ..., as the light walks over the tomatoes, It keeps shining. It didn't darken. Why? Because the tomatoes are ripe, and light is reflected inside the tomatoes, ... |
| DeepL Translate | ..., as the light traveled over the tomatoes, it kept shining. It doesn't dim. Why? Because the tomatoes are ripe and the light is reflecting inside the tomatoes, ... |
| Baseline Model (Sentence-level) | ..., as the light goes over the tomato, It's always glowing. It's not darkening. Why? Because the tomato is ripe, and light is reflected inside the tomato, ... |
| Baseline Model (Document-level) | ..., as the light passes over the tomato, It keeps flashing. It doesn't get darker. Why? Because the tomatoes are ripe , and the light is is reflected inside the tomato, ... |
| TRIP | ..., as the light passes over the tomato, It's flashing all the time. It's not getting darker. Why? Because the tomato is ripe, and the light is reflected inside the tomato, ... |
| | Case 3: Conjunction Presence (Xiong et al., 2019; Sun et al., 2022) |
| Source | ……，我想提醒大家，我已经谈论了很多前人的事情。我还想考虑一下，民主会是什么样子,或者是已经是什么样子的可能性如果我们可以让更多的母亲参与进来，…… |
| Reference | ..., I want to suggest to you that I've been talking a lot about the fathers. And I want to think about the possibilities of what democracy might look like, or might have looked like, if we had more involved the mothers, ... |
| Google Translate | ..., I want to remind everyone that I've talked a lot about my predecessors. I also want to think about what democracy would look like, or is it already What the possibilities look like if we could get more mothers involved, ... |
| Microsoft Translator | ..., I want to remind you that I have talked a lot about my predecessors. I would also like to consider what democracy would look like, or already be What kind of possibilities if we can involve more mothers , ... |
| DeepL Translate | ..., I want to remind you that I've talked about a lot of things that have come before. I also want to consider the possibility of what democracy would look like, or what it already looks like if we could get more mothers involved in, ... |
| Baseline Model (Sentence-level) | ..., I want to remind you that I've talked about a lot of my predecessors. I also want to think about what democracy might look like if we could get more mothers involved, ... |
| Baseline Model (Document-level) | ..., I'd like to remind you that I've talked about a lot of things before. I'd also like to think about the possibilities of what democracy might look like, or what it might be like, if we could get more mothers to participate, ... |
| TRIP | ..., I want to remind you that I've talked a lot about the past. And I want to think about the possibilities of what democracy might look like, or already looks like, if we can get more mothers involved, ... |

Table 5: Cases from TED Talks demonstrate that TRIP captures better tense consistency, noun-related issues, and conjunction presence. We highlight the correct translation in aqua (the darker one when printed in B&W), and the mistakes in hot pink (the lighter one when printed in B&W). Google Translate: https://translate.google.com/, Microsoft Translator: https://www.bing.com/translator, DeepL Translate: https://www.deepl.com/translator. Time-stamped on 15th June 2023, can be subject to change.

## 3.3 Cross-lingual Abstractive Summarization

**Experimental Settings** We follow the same setting used by DOCmT5 (Lee et al., 2022) to evaluate cross-lingual abstractive summarization on the benchmark of Wikilingua (Ladhak et al., 2020). The only difference is that they put a special prefix *"Summarize X to Y"* where X and Y are the source and target language tags for summarization like mT5. We put a target language tag as the prefix. We use the F1 measure for ROUGE-1/ROUGE-2/ROUGE-L scores (Lin, 2004) for evaluation.

**Baseline Systems** We report the scores for DOCmT5 taken from Lee et al. (2022), and we use prior SOTA scores from the official GEM benchmark (Gehrmann et al., 2021) for mT5, ByT5 (Xue

et al., 2022). We also employ mBART and the Baseline Models as the baselines. See Appendix C for the number of model parameters.

**Results** Table 6 demonstrates that TRIP clearly exceeds previous SOTA systems in several directions, including up to 8.9 ROUGE-L points in (Hi → En) compared to DOCmT5. Hence, we conclude that TRIP is an effective multilingual pre-training framework for cross-lingual abstractive summarization. We postulate that the improvement is attributed to the trilingual pre-training objective overlooked by previous works such as DOCmT5.

Also, we found that using bilingual data for Baseline Model+ seems less beneficial on Wikilingua for cross-lingual abstractive summarization. TRIP

| Model | Tr→En | Vi→En | Ru→En | Es→En | Hi→En | Fr→En | Id→En |
|---|---|---|---|---|---|---|---|
| mT5-XL† | 40.0/18.3/33.3 | 37.6/14.9/31.2 | **37.2/14.6/30.9** | 41.2/17.2/34.6 | -/-/- | -/-/- | -/-/- |
| ByT5† | 35.9/15.8/29.8 | 32.7/12.2/27.2 | 31.4/11.0/26.2 | 35.1/13.5/29.1 | -/-/- | -/-/- | -/-/- |
| mBART† | 34.4/13.0/28.1 | 32.0/11.1/26.4 | 33.1/11.0/27.8 | 38.3/15.4/32.4 | -/-/- | -/-/- | -/-/- |
| DOC-mT5† | 37.7/16.7/31.4 | 32.4/11.9/27.0 | 33.6/12.8/28.5 | 36.8/15.0/31.5 | 34.2/13.3/27.9 | 36.3/14.3/30.8 | 35.2/13.7/29.5 |
| Baseline Model | 42.4/19.7/36.4 | 38.5/15.8/32.9 | 34.9/13.4/29.7 | 36.9/14.8/31.4 | 40.9/18.0/35.0 | 37.3/14.9/31.9 | 37.8/15.3/32.2 |
| Baseline Model⁺ | 42.6/19.6/36.6 | 38.8/16.1/33.1 | 34.9/13.3/29.6 | 37.1/14.8/31.5 | 40.7/18.1/34.9 | 37.2/14.9/31.8 | 37.6/15.2/31.9 |
| **TRIP (Ours)** | **45.3/22.5/39.0** | **40.8/17.3/34.4** | 36.6/14.6/30.8 | 38.7/15.9/32.7 | **42.8/19.9/36.8** | **38.5/16.0/32.9** | **39.4/16.4/33.3** |

Table 6: Results for cross-lingual abstractive summarization on Wikilingua in (X → En). We report the scores of F-measure for ROUGE-1/ROUGE-2/ROUGE-L. -: the score is not reported. †: the scores are taken from Lee et al. (2022) and the official GEM benchmark (Gehrmann et al., 2021): https://gem-benchmark.com/results.

| Model | Fr→En | De→En | Zh→En | Vi→En | Cs→En | Th→En | Avg. |
|---|---|---|---|---|---|---|---|
| ChatGPT | 40.47 | 36.76 | 23.31 | 28.26 | 30.29 | 20.94 | 30.01 |
| **TRIP** | **54.13** | **49.94** | **28.45** | **41.19** | **42.73** | **34.92** | **42.00** |

Table 7: Comparison of TRIP to ChatGPT on the task of document-level machine translation on TED Talks in the direction of (X → En). The results are snapshotted in May 2023 and can be subject to change.

clearly surpasses the Baseline Model⁺. This observation indicates that the trilingual objective with the Grafting mechanism is superior to the conventional bilingual objectives for multilingual pre-training.

**Case Study** Table 8 in Appendix shows three case studies that TRIP outputs better abstractive cross-lingual summarization. For space reasons, we leave more details in Appendix B.

## 4 Related Work

### 4.1 Multilingual Pre-training

Multilingual pre-training has achieved great success. Previous works can be categorized into two streams: monolingual pre-training (Conneau et al., 2020; Liu et al., 2020; Xue et al., 2021) and bilingual pre-training (Huang et al., 2019; Chi et al., 2021; Ouyang et al., 2021; Tang et al., 2021; Chi et al., 2021; Reid and Artetxe, 2022; Lee et al., 2022). Monolingual pre-training uses monolingual corpora in many different languages and perturbs the inputs with sentence permutation (Liu et al., 2020) and span corruption (Xue et al., 2021) and requires the model to reconstruct the original input. Reid and Artetxe (2022) also proposes dictionary denoising on monolingual data. For bilingual pre-training, Tang et al. (2021) uses clean sentence-level bilingual translation pairs on pre-trained models to improve MT. Chi et al. (2021) extends mT5 with objectives such as translation span corruption. DOCmT5 (Lee et al., 2022) creates synthetic bilingual translation pairs and uses sentence permuta-

tion for a document-level multilingual pre-training.

### 4.2 Document-level Cross-lingual Tasks

Document-level MT and cross-lingual abstractive summarization are the two document-level cross-lingual tasks that we investigate in this paper.

Document-level MT (Miculicich et al., 2018; Maruf et al., 2019, 2021; Lu et al., 2022) is a challenging translation task, possibly due to the long input problem (Pouget-Abadie et al., 2014; Koehn and Knowles, 2017) when directly modelling the long document and the necessity in understanding contexts (Voita et al., 2018, 2019). Therefore, many works focus on using sentence-level models with a smaller contextual window to simulate document-level MT (Zheng et al., 2020; Chen et al., 2020). This paper follows the challenging setting (Bao et al., 2021; Lee et al., 2022) that directly optimizes a document-level model with a longer context window that provides a richer source of context, which is also a double-edged sword that could be harder due to the long input problem.

Abstractive summarization is a generation task that requires an understanding of texts (Chopra et al., 2016; Fan et al., 2018). We focus on a cross-lingual setting where source and target are written in different languages (Ladhak et al., 2020).

## 5 Conclusions

We present a novel sequence-to-sequence multilingual document-level pre-training methodology called **TRIP**, which is the first in our field to propose a trilingual objective for multilingual pre-training through Grafting. We also propose a novel method to construct high-quality trilingual document pairs. Experimental results indicate that TRIP achieves competitive SOTA scores on both multilingual document-level machine translation and cross-lingual abstractive summarization. Future works could improve TRIP to include polygonal parallel

translation pairs in multilingual pre-training. We plan to release the model checkpoints and a manually annotated benchmark created using our created document-level corpus MTDD to facilitate future research on multilingual document-level MT.

## Limitations

**TRIP** TRIP leverages high-quality document-level trilingual translation pairs for pre-training on multilingual models. It is usually harder to collect high-quality trilingual data than to collect monolingual data written in different languages used by conventional methods. While we can possibly relax the quality bar for the data, additional experiments should be done to verify this view.

**MTDD** We create MTDD, a corpus that is composed of trilingual document pairs. It could be further extended to include polygonal parallel document pairs to provide a stronger signal for multilingual pre-training. We leave this to future works.

**Large Language Models** Large language models (LLMs) such as ChatGPT have shown good translation abilities (Lu et al., 2023), while they still lag behind supervised systems (Jiao et al., 2023; Zhu et al., 2023). We conduct a limited comparison to them, as they are much larger in their number of parameters than the systems described in this work.

## Ethics Statement

We honour and support the EMNLP Code of Ethics. The datasets used in this work are well-known and widely used, and the dataset pre-processing does not use any external textual resource. We also curate a corpus for pre-training language models. Although we have made our best efforts in reducing potentially offensive and toxic data, the models are subject to generating offensive context. But the issues mentioned above are widely known to exist for these models commonly. Any content generated does not reflect the view of the authors.

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

| | *Case 1* |
|---|---|
| **Source** | Ayakkabılarını (ve bağcıklarınla tabanlıklarını) kuruması için orta derecede ışık alan bir yere koy. Sıcak bir yere (örneğin, radyatörün yanına) ya da doğrudan güneş ışığına koyma çünkü bu, ayakkabılara zarar verebilir. Ayakkabılarını kurutucuya koymak tavsiye edilmez çünkü kurutucu, ayakkabı tabanlarını yamultabilir. |
| **Source (Google-translated)** | Put your shoes (and your laces and insoles) in a moderately light place to dry. Do not place it in a hot place (for example, near a radiator) or in direct sunlight as this may damage the shoes. Putting your shoes in the dryer is not recommended because the dryer can warp the soles of your shoes. |
| **Reference** | Air-dry your shoes. |
| **Baseline Model (Document-level)** | Allow your shoes (and laces) to dry. |
| **TRIP** | Let your shoes (and the laces) air dry. |
| | *Case 2* |
| **Source** | Bunun için yeşil bir arka plan üzerindeki beyaz konuşma balonuna dokun. Ana Ekranlarından birinde olması gerekir. 'ye dokun. Mesajlar ekranının sol üst köşesindedir. Açık bir sohbetin varsa Mesajlar menüsüne dönmek için ekranın sol üst köşesindeki < butonuna dokun. 'e dokun. Ekranının sağ alt köşesindedir. Seçilen mesajların silinir. |
| **Source (Google-translated)** | To do this, tap the white speech bubble on a green background. It should be on one of their Home Screens. Tap It's in the upper-left corner of the Messages screen. If you have an open chat, tap the < button in the upper left corner of the screen to return to the Messages menu. Tap . It's in the lower right corner of your screen. The selected messages are deleted. |
| **Reference** | Open your iPhone's messages. Tap Edit. Select each conversation you wish to delete. Tap Delete. |
| **Baseline Model (Document-level)** | Open your iPhone's Settings . Tap Messages. Tap Delete Messages. |
| **TRIP** | Open Messages. Tap the Messages tab. Tap Delete. Tap Delete to confirm. |
| | *Case 3* |
| **Source** | Bazıları için geçmiş yaşamlar gidilecek bir yer değil, seni sen yapan şeyin bir kısmıdır. İnsanlığın tarihi boyunca birçok kültür reenkarnasyonu inançlarının merkezine koymuştur. İslam ve Hıristiyanlık reenkarnasyona inanmasa da, Hinduistler, bazı Museviler ve bazı Budistler buna inanır. En iyisi kendini bir dine tümüyle adamaktansa (çünkü dinler çok kısıtlayıcı olabilir) kendi yolunu keşfetmen. Kendi manevi doğrularını kendin bul. |
| **Source (Google-translated)** | For some, past lives are not a place to go but part of what makes you who you are. Throughout the history of humanity, many cultures have put reincarnation at the center of their beliefs. Although Islam and Christianity do not believe in reincarnation, Hinduists, some Jews, and some Buddhists do. It's best to explore your own path rather than devote yourself entirely to a religion (because religions can be too restrictive). Find your own spiritual truth. |
| **Reference** | Become spiritual. |
| **Baseline Model (Document-level)** | Understand that some people believe in reincarnation. Find your own way. |
| **TRIP** | Explore your own spiritual journey. |

Table 8: Three case studies from Wikilingua (Tr → En) demonstrate that TRIP outputs better summarization.

| | Zh | Cs | Fr | Vi | Th | De |
|---|---|---|---|---|---|---|
| Zh | - | 17.07/16.96 | 26.78/25.58 | 26.66/27.26 | 16.99/16.65 | 21.85/21.46 |
| Cs | 28.60/27.32 | - | 34.69/34.06 | 31.33/31.72 | 18.72/18.40 | 30.84/30.17 |
| En | 31.07/31.33 | 30.94/30.74 | 49.58/49.22 | 38.23/37.91 | 22.19/21.92 | 38.63/38.50 |
| Fr | 28.90/27.55 | 26.47/24.83 | - | 35.05/35.09 | 19.66/19.93 | 33.86/33.47 |
| Vi | 26.93/27.01 | 21.08/20.77 | 34.37/34.37 | - | 18.94/18.46 | 26.13/25.44 |
| Th | 25.56/24.81 | 18.24/17.74 | 27.74/26.54 | 27.69/27.38 | - | 22.78/21.87 |
| De | 27.91/27.72 | 24.72/24.63 | 37.03/36.87 | 30.70/31.10 | 19.61/19.37 | - |

Figure 4: Results on TED Talks in (X→X) with our TRIP checkpoint pre-trained in (X→En) directions only. The scores are written in TRIP as the former and the Baseline Model as the latter. Rows represent the source languages and columns represent the target languages. We highlight in aqua when TRIP wins (darker one when printed in B&W) and in hot pink (lighter one when printed in B&W) when the Baseline Model wins.

## A Unseen (X→X) Language Pairs on MT

Figure 4 reports the performance on TED Talks in the direction of (X→X) with our TRIP checkpoint pre-trained in (X→En) directions with our corpus. The row represents the translation source language and the column represents the translation target language. TRIP clearly improves most of these translation directions which are unseen during pre-training. This indicates that fact the TRIP can generalize the cross-lingual understanding ability to unseen language pairs. This aligns with the fact reported in Lee et al. (2022).

## B Case Study on Summarization

Table 8 shows that TRIP outputs better summarization in (i) precisely capturing the context in Case 1, (ii) outputting consistent nouns, i.e., 'messages' instead of 'settings' in Case 2 and (iii) producing concise and accurate summarization in Case 3. This highlights that TRIP captures better cross-lingual understanding than the baseline system, which effectively mitigates cross-lingual ambiguities.

## C Number of Model Parameters

| Model | Number of Parameters |
|---|---|
| M2M-100 | 418M |
| mBART | 611M |
| MARGE | 963M |
| mT5 | 1.23B* |
| DOCmT5 | 1.23B* |
| ByT5-Small | 300M |
| ByT5-Base | 582M |
| ByT5-Large | 1.23B* |
| mT5-XL | 3.74B |
| Baseline Model | 862M |
| Baseline Model+ | 862M |
| **TRIP (Ours)** | 862M |

Table 9: Comparison in the number of parameters for the pre-trained models used in our experiments. ∗: these models all use the model architecture of mT5-Large, and we report the number of model parameters taken from the original paper of mT5 reported by Xue et al. (2021).

| Source | Target | Size/GB | Source | Target | Size/GB |
|---|---|---|---|---|---|
| Es | En | 3.22 | Pt | Es | 2.71 |
| Es | Ca | 2.07 | Uk | Ru | 1.60 |
| Fr | Es | 1.47 | Es | Pt | 1.47 |
| En | De | 1.25 | Pt | En | 1.14 |
| Ca | Es | 1.12 | Fr | En | 1.03 |
| Ru | Uk | 0.87 | Pt | Fr | 0.73 |

Table 10: A language list in ISO code for the top 12 language directions for the bilingual high-quality pre-training data to illustrate the scale of size.

Table 9 presents the number of model parameters for the pre-trained models used in our experiments.

For the scores of ByT5 presented in Table 6, we report the maximum scores for each direction among ByT5-Base, ByT5-Small, and ByT5-Large. This is due to space reasons. See https://gem-benchmark.com/results for the tailored scores.

## D MTDD Corpus Scale

Table 10 presents the top-12 English-centric bilingual data statistics to illustrate the scale of MTDD. The total size of the data is about 40/80 GB respectively for the bilingual and the trilingual data applied with Grafting.