# OpenReview forum: "TRIP: Accelerating Document-level Multilingual Pre-training via Triangular Document-level Pre-training on Parallel Data Triplets"
_EMNLP/2023/Conference — EMNLP 2023 Findings_

### Official Review · Reviewer_fb6A · 2023-07-28

**Soundness:** 3

**Excitement:**

3: Ambivalent: It has merits (e.g., it reports state-of-the-art results, the idea is nice), but there are key weaknesses (e.g., it describes incremental work), and it can significantly benefit from another round of revision. However, I won't object to accepting it if my co-reviewers champion it.

**Paper Topic And Main Contributions:**

This paper proposes a novel pretraining technique that only requires document-level alignment for three (or more) languages (a dataset of which they also release). Given a three-way correspondence between documents across 3 languages, for all three language pairs their method a) combines half of a document from one language with another ("grafting"), b) permutes the result, c) adds span corruption. The remaining unchanged document is used as the reference. Their method is robust to noisy sentence alignments, and they show non-trivial improvements over a strong document-level baseline (which is trained on the bilingual document pair data from the dataset) in translation and cross-lingual abstractive summarization. The authors also leverage prior work (ParaCrawl) to extract bilingual document pairs from which they extract trilingual matches, and release this dataset under the name MTDD.

**Questions For The Authors:**

- any thoughts on why performance improvements seem much better on TEDTalks vs. other corpora?

**Reasons To Accept:**

- dataset with trilingual document pairs
- pretraining technique with relatively minimal requirements (i.e. can be applied on other datasets that fit the criteria relatively easily)
- strong baselines and good to evaluate on two both xlingual summarization and MT.

**Reasons To Reject:**

- there is a claim and a footnote on robustness to documents that aren't cleanly aligned at the sentence-level. I think this is an important claim, and could be backed up with more evidence (maybe I missed it, but simply presenting the result in the appendix would suffice)
- lack of analysis on how performance is heterogeneous across different language pairs. Also, the motivation of the paper was interesting in that grafting can help bridge or transfer linguistic phenomena, but there wasn't much analysis done on this.

**Reproducibility:**

4: Could mostly reproduce the results, but there may be some variation because of sample variance or minor variations in their interpretation of the protocol or method.

**Reviewer Confidence:**

3: Pretty sure, but there's a chance I missed something. Although I have a good feel for this area in general, I did not carefully check the paper's details, e.g., the math, experimental design, or novelty.

---

> ### Author Rebuttal · Authors · 2023-08-23
>
> Dear reviewer,
>
> Thank you for spending time assessing the paper and writing the reviews.
>
> There might be some miscommunication and we would like to provide our response to your concerns.
>
> > **Q1:** there is a claim and a footnote on robustness to documents that aren't cleanly aligned at the sentence-level. I think this is an important claim, and could be backed up with more evidence (maybe I missed it, but simply presenting the result in the appendix would suffice)
>
> **A1:** Here are two points we would like to address. (i) We originally supported such a claim via our empirical observation described in the paper that TRIP can obviously improve performance on MT and Cross-lingual Summarisation. (ii) Here is another shortcut to further simplify and solve your concern. Incorporating noise has been previously shown as useful for pre-training (e.g., sentence order and masked words in BART). Following this solid statement, the alignment (sentence order & presence of the words, or, simply masked words) is also not malicious to TRIP. This is then verified by our experiments.
>
> > **Q2:** lack of analysis on how performance is heterogeneous across different language pairs. Also, the motivation of the paper was interesting in that grafting can help bridge or transfer linguistic phenomena, but there wasn't much analysis done on this.
>
> **A2:** We will add more analysis on how performance is heterogeneous across different language pairs in our succeeding work.
>
> Regarding the linguistic phenomena, instead of human evaluation, we have used an automatic evaluation (BlonDe) which is a good evaluation metric for document-level MT that considers various consistency issues. There is also a case study in Table 5 that shows TRIP effectively improves various linguistic considerations.
>
> > **Q3:** any thoughts on why performance improvements seem much better on TEDTalks vs. other corpora?
>
> **A3:** One clue could be our experimental setup. We have the Baseline Model and Baseline Model+ as our baseline, where they can have different performance standards on different datasets. This can affect the performance gain from TRIP. Another clue is that we are also motivated by the linguistic phenomena as discussed in your last comments. We postulate that such motivation can bring better benefits to the talk text style on TEDTalks than the other two datasets on MT.
>
> Best Regards,
>
> Authors

---

### Official Review · Reviewer_4LSK · 2023-08-05

**Soundness:** 3

**Excitement:**

3: Ambivalent: It has merits (e.g., it reports state-of-the-art results, the idea is nice), but there are key weaknesses (e.g., it describes incremental work), and it can significantly benefit from another round of revision. However, I won't object to accepting it if my co-reviewers champion it.

**Paper Topic And Main Contributions:**

This paper proposes to adopt trilingual document-level data to improve multilingual document-level pretraining. A trilingual example is a triplet, including three inputs from three different languages which are translations of each other. To get such data, the authors collected MTDD, a large-scale multilingual document-level data covering 67 languages. Experiments on several translation tasks and cross-lingual tasks show improved performance.

**Questions For The Authors:**

* As one main contribution, the description of MTDD is vague. Section D is not enough to illustrate the statistics for MTDD. Readers would expect a table showing the number of sentences/documents/tokens for each triplet language rather than only giving some vauge numbers for top-12 language pairs. For example, some language pairs might only have a handful of examples, which are not significant at all.
* The experimental settings are pretty unclear with respect to initialization, pretraining and finetuning. The lack of these details hinders the readers from fully understanding the effectiveness of the proposed trilingual pertaining.
  - What's the Baseline Model? How did you train it? What hyperparameters and training data did you use for it?
  - How did you train Baseline Model +? Did you apply translation objective or denoising training on MTDD?
  - Did you perform finetuning on each test task? How did you perform the finetuning? Did you also apply finetuning to similar finetuning to the baselines, such as Baseline Model+, mBart, and m2m-100?
  - Is your TRIP model a multilingual model? How did you mix different languages? Why would you prefer Yang et al. (2021)'s sentencepiece model for your setup as you have 67 languages, many of which shouldn't be covered?
  - Many translation tasks are essentially bilingual tasks. In the results, TRIP almost always outperforms Baseline Model+. Could you explain why the trilingual objective would help a bilingual task?
* The title shows "accelerating ... pretraining". I didn't find any evidence from the paper supporting the acceleration statement.
* M2M-100 adopts many bilingual and monolingual data for the pretraining, but the authors compared it with a significantly smaller one in the paper. I believe the comparison is unfair. What about the 1B M2M-100 model?
* Scaling model size has been shown effective for the pretraining. Would scaling still prefer the trilingual objective over the bilingual one?

I'd love to increase my score if the authors could address my concerns.

== Update
The authors' response addressed some of my concerns and I increased the score accordingly.

**Reasons To Accept:**

1. MTDD, a large-scale multilingual and trilingual document-level corpus.
2. Trilingual pretraining that adopts triplet data to improve pretraining

**Reasons To Reject:**

1. Details of MTDD are unclear, particularly the statistic and distribution for different language pairs.
2. Experimental settings are unclear: not sure how the authors performed the pretraining.
3. Comparisons might be misleading. The conclusions -- the value of MTDD and trilingual training -- will need further evidence.

**Reproducibility:**

3: Could reproduce the results with some difficulty. The settings of parameters are underspecified or subjectively determined; the training/evaluation data are not widely available.

**Reviewer Confidence:**

4: Quite sure. I tried to check the important points carefully. It's unlikely, though conceivable, that I missed something that should affect my ratings.

---

> ### Author Rebuttal · Authors · 2023-08-23
>
> Dear reviewer,
>
> Thank you for spending time assessing the paper and writing the reviews.
>
> There might be some miscommunication and we would like to provide our response to your concerns.
>
> >**Q1:** The experimental settings are pretty unclear with respect to initialization, pretraining and finetuning. The lack of these details hinders the readers from fully understanding the effectiveness of the proposed trilingual pertaining.
>
> **A1:** **We assure you that we have conducted the experiments fairly. More experimental settings are given below. We will add them to our camera-ready version.**
>
> > **Q2:** What's the Baseline Model? How did you train it? What hyperparameters and training data did you use for it?
>
> **A2:** The Baseline Model is trained exactly following Yang et al. (2021). (Line 208), using the exactly same hyperparameters and training data. One may refer to their paper for detailed settings.
>
> > **Q3:** How did you train Baseline Model +? Did you apply translation objectives or denoising training on MTDD?
>
> **A3:** Yes, we apply shuffling and masking on MTDD (bilingual version) to Baseline Model+. TRIP uses shuffling and masking on the trilingual version of MTDD.
>
> > **Q4:** Did you perform finetuning on each test task? How did you perform the finetuning? Did you also apply finetuning to similar finetuning to the baselines, such as Baseline Model+, mBart, and m2m-100?
>
> **A4:** We apply fine-tuning fairly on all baselines. We apply the fine-tuning following hyperparameters in lines 227-238. We do not have Grafting here, and we apply validation patience of 5 epochs.
>
> > **Q5:** Is your TRIP model a multilingual model? How did you mix different languages? Why would you prefer Yang et al. (2021)'s sentencepiece model for your setup as you have 67 languages, many of which shouldn't be covered?
>
> **A5:** TRIP is a multilingual model. One reason to use the model from Yang et al. (2021) is for the scalability of future development based on our current checkpoint. And there is a plan to cover hundreds of languages in a future version of TRIP.
>
> > **Q6:** Many translation tasks are essentially bilingual tasks. In the results, TRIP almost always outperforms Baseline Model+. Could you explain why the trilingual objective would help a bilingual task?
>
> **A6:** The trilingual objective incorporates more information into the pre-training stage. This helps in resolving coherence issues. We present in Figure 1 for our motivation and Table 5 for the showcases.
>
> > **Q7:** The title shows "accelerating ... pretraining". I didn't find any evidence from the paper supporting the acceleration statement.
>
> **A7:** No, we are not speeding up the model in any format (efficiency, decoding). We originally intended to say that we are "accelerating" the progress of the multilingual pre-training in its literature. We chose the word accelerating to make it verbally catchy. We are open to replacing it with other words like "improving" or "advancing".
>
> > **Q8:** M2M-100 adopts many bilingual and monolingual data for the pretraining, but the authors compared it with a significantly smaller one in the paper. I believe the comparison is unfair. What about the 1B M2M-100 model?
>
> **A8:** We note that M2M-100 uses different data for pre-training. To conduct a fair comparison, our original intention is to focus more on the (Baseline Model+ VS TRIP). Doing trilingual pre-training with TRIP clearly surpasses Baseline Model+, which supports our claim about the effectiveness of TRIP. Nevertheless, according to the original paper of M2M-100 (Figure 7 in their paper), the performance gap is not large between 418M and 1B (2 points of BLEU).  However, TRIP obviously surpasses M2M-100 (418M) by 5-10 points of BLEU (Table 2, 3). Here are some additional results:
>
> Document-level M2M-100 on TEDTalks benchmark (Table 2)
> Model | Fr→En | De→En | Zh→En | Vi→En | Cs→En | Th→En | Avg.
>  :----: |  :----: |  :----: |  :----: |  :----: |  :----: |  :----: |  :----:
> M2M (418M) | 49.43 | 43.82 | 26.63 | 35.91 | 39.04 | 25.93 | 36.79
> M2M (1B__)  | 51.13 | 46.01 | 27.66 | 37.82 | 40.88 | 27.33 | 38.47
> TRIP (Ours)  |  **51.94** | **48.24** | **31.63** | **40.52** | **44.22** | **32.87** | **41.57**
>
> Document-level M2M-100 on News benchmark (Table 3)
> Model | Fr→En | De→En | Zh→En | Cs →En | Avg.
>  :----: |  :----: |  :----: |  :----: |  :----: |  :----:
> M2M (418M) | 32.67 | 25.78 | 17.85 | 29.06 | 26.34
> M2M (1B__)  | 34.81 | 27.25 | 20.01 | 30.92 | 28.25
> TRIP (Ours)  | **39.49** | **35.48** | **27.58** | **38.06** | **35.15**
>
> > **Q9:** Scaling model size has been shown effective for the pretraining. Would scaling still prefer the trilingual objective over the bilingual one?
>
> **A9:** We plan to investigate it in our future work and we will open-resource TRIP model checkpoints with various sizes. Pre-training can be computationally expensive, and we are trying our best to analyse other model sizes as well as scaling to even larger model sizes.
>
> > **Q10:** As one main contribution, the description of MTDD is vague. Section D is not enough to illustrate the statistics for MTDD. Readers would expect a table showing the number of sentences/documents/tokens for each triplet language rather than only giving some vague numbers for top-12 language pairs. For example, some language pairs might only have a handful of examples, which are not significant at all.
>
> **A10:** To get a better insight into the dataset statistics, for example, we report in our submission that we have Es-En, about 3.22GB 250 million English tokens on MTDD. This converts to tens of millions of English sentences and millions of documents. We will release all information/statistics with a camera-ready version of this work as soon as possible. We will also release more details, e.g., more information on data sources upon publication. We did not write much in this submission to avoid a potential violation of anonymity.
>
> > I'd love to increase my score if the authors could address my concerns.
>
> We hope our response can address your concerns and we appreciate it if you could kindly raise your scores. Thank you very much.
>
> Best Regards,
>
> Authors

---

### Official Review · Reviewer_WyK4 · 2023-08-06

**Soundness:** 4

**Excitement:**

4: Strong: This paper deepens the understanding of some phenomenon or lowers the barriers to an existing research direction.

**Justification For Ethical Concerns:**

NO ethical concerns

**Paper Topic And Main Contributions:**

The paper is well motivated, by mix-and-mask bilingual comparable docs at sentence level.  One motivation is to fix the tense and gender issues in hallucinated translations or summarizations. The languages chosen, zh, ja and en, are suitable for such study.   However, the experiments were not detailed on simple qualitatively error analysis on these two streams.  This maybe the biggest key stats lacking from experiments.

Paper is well written, and it'd be great to release MTDD corpus for the community; the authors also claimed they will release the checkpoint w/ code for public research baseline.

**Questions For The Authors:**

There are also pro-drops for Zh, esp. in social media style languages, as shown in Figure-1.

Besides Gender and Tense, can the authors do more analysis on pro-drops for pronouns AND SVO(zh/en) vs VSO(ja) in translations error analysis?

**Reasons To Accept:**

Idea is simple and convincing with all experiments;  paper is well written, and the authors claimed they will release the checkpoint w/ code for public research baseline on MTDD; the research community will benefit from this.

**Reasons To Reject:**

One motivation is to fix the tense and gender issues in hallucinated translations or summarizations. The languages chosen, zh, ja and en, are suitable for such study.   However, the experiments were not detailed on simple qualitatively error analysis on these two streams.

**Reproducibility:**

3: Could reproduce the results with some difficulty. The settings of parameters are underspecified or subjectively determined; the training/evaluation data are not widely available.

**Reviewer Confidence:**

4: Quite sure. I tried to check the important points carefully. It's unlikely, though conceivable, that I missed something that should affect my ratings.

---

> ### Author Rebuttal · Authors · 2023-08-23
>
> Dear reviewer,
>
> Thank you for spending time assessing the paper and writing the reviews.
>
> We would like to provide our response to your concerns.
>
> > **Q1:** One motivation is to fix the tense and gender issues in hallucinated translations or summarizations. The languages chosen, zh, ja and en, are suitable for such study. However, the experiments were not detailed on simple qualitative error analysis on these two streams.
>
> **A1:** A showcase in Table 5 (Case 1) demonstrates that TRIP mitigates the tense issue. TRIP also clearly enhances BlonDe scores (refer to Figure 3), and BlonDe primarily focuses on assessing the coherence of tense and noun translations at the document level.
>
> > **Q2:** There are also pro-drops for Zh, esp. in social media-style languages, as shown in Figure-1.
> Besides Gender and Tense, can the authors do more analysis on pro-drops for pronouns AND SVO (zh/en) vs VSO (ja) in translation error analysis?
>
> **A2:** Thank you for the excellent suggestion. We will include an analysis of pro-drop in our future endeavours.
>
> Best Regards,
>
> Authors

---

### Meta-Review · Area_Chair_AcJ4 · 2023-09-18

**Recommendation:** 3

**Metareview:**

The paper received mixed reviews with respect to soundness and excitement.

While the MTDD corpus has been praised as an interesting resource for the community, the reviewers have raised concerns about the evaluation protocol and the validity of some claims.

---

### Decision · Program_Chairs · 2023-10-07

**Decision:**

Accept-Findings

**Comment:**

The paper received mixed reviews with respect to soundness and excitement.

While the MTDD corpus has been praised as an interesting resource for the community, the reviewers have raised concerns about the evaluation protocol and the validity of some claims.